# Screening Pregnant Women for Bacterial Vaginosis Using a Point-of-Care Test: A Prospective Validation Study

**DOI:** 10.3390/jcm10112275

**Published:** 2021-05-24

**Authors:** Philipp Foessleitner, Herbert Kiss, Julia Deinsberger, Julia Ott, Lorenz Zierhut, Klara Rosta, Veronica Falcone, Alex Farr

**Affiliations:** 1Department of Obstetrics and Gynecology, Division of Obstetrics and Feto-Maternal Medicine, Medical University of Vienna, A-1090 Vienna, Austria; philipp.foessleitner@meduniwien.ac.at (P.F.); herbert.kiss@meduniwien.ac.at (H.K.); n1547087@students.meduniwien.ac.at (J.O.); lorenz.zierhut@gmail.com (L.Z.); klara.rosta@meduniwien.ac.at (K.R.); veronica.falcone@meduniwien.ac.at (V.F.); 2Department of Dermatology, Skin and Endothelium Research Division (SERD), Medical University of Vienna, A-1090 Vienna, Austria; julia.deinsberger@meduniwien.ac.at

**Keywords:** antenatal care, *Gardnerella vaginalis*, point-of-care test, pregnancy, bacterial vaginosis, preterm birth, infant mortality

## Abstract

Bacterial vaginosis in early pregnancy is associated with an increased risk of preterm birth. The introduction of a simple screen-and-treat program into antenatal care was shown to significantly reduce the rate of preterm birth. The gold standard for diagnosing bacterial vaginosis is Gram staining, which is, however, time-consuming and requires laboratory facilities. The objective of this prospective study was to validate a point-of-care sialidase activity detection test (*OSOM^®^ BVBLUE^®^ Test*) for asymptomatic pregnant women and evaluate its accuracy as a screening tool. We enrolled 200 pregnant participants, 100 with Gram staining-confirmed bacterial vaginosis and 100 healthy controls. Compared to Gram staining, the point-of-care test showed a sensitivity of 81%, specificity of 100%, positive predictive value of 100%, and negative predictive value of 98.1%. In conclusion, we found that the *OSOM^®^ BVBLUE^®^ Test* was an accurate method for diagnosing bacterial vaginosis in asymptomatic pregnant women. This point-of-care test can therefore be considered a reliable and easy-to-use screening tool for bacterial vaginosis during pregnancy.

## 1. Introduction

Early pregnancy bacterial vaginosis (BV) has been associated with spontaneous preterm birth (PTB), defined as birth before 37 weeks of gestation [1,2]. This is of particular interest since PTB is still the main cause of perinatal morbidity and mortality in industrialized countries [3]. Numerous causes increase the risk for this multifactorial event, including a high maternal body mass index, nicotine use, advanced maternal age, various diseases, previous PTB, and vaginal infections [4,5]. In fact, maternal infections during pregnancy account for up to 40% of PTB cases [6]. Vaginal dysbiosis and consecutive BV increase the risk of PTB, which is of particular interest since its incidence is relatively high among women of reproductive age [2,7].

Vaginal dysbiosis might progress to BV by a decrease in the H_2_O_2_-producing lactobacilli concentration and an increase in the abundance of anaerobic bacteria, especially *Gardnerella vaginalis, Prevotella bivia,* and *Atopobium vaginae* [8,9]. These microorganisms produce the enzyme sialidase that causes the release of sialic acid [10,11]. Sialic acid is used by these bacteria to adhere to cellular and inert surfaces as sources of nutrition and to modify the immune response and normal mucus barrier [11]. High levels of sialidase are strongly associated with early spontaneous PTB and late miscarriage [12].

Timely diagnosis and treatment of BV during early pregnancy significantly reduces the rate of PTB [2]. Our previous work demonstrated an impressive reduction in PTB by introducing a simple screen-and-treat program for asymptomatic pregnant women in early gestation [7,13]. In clinical practice, the Amsel criteria [14] were shown to be inadequate for BV diagnosis due to their low sensitivity of only 50% [15]. The use of bacterial culture methods during routine care is inefficient because of the relatively high costs and time latency to receipt of the report. Moreover, culture methods do not allow for clear identification of pathogenic strains among the bacterial variety in the vaginal ecosystem. For this reason, microscopic evaluation of Gram-stained smears is widely considered the gold standard method for BV diagnosis [16].

Alternative diagnostic tools to Gram staining, which requires laboratory facilities and trained and experienced staff, are warranted. A possible approach is the use of point-of-care tests that promise accurate and rapid diagnosis [15,17,18]. These tests can be easily performed without additional equipment or staff and provide results within just a few minutes [18]. One of these tests, the *OSOM^®^ BVBLUE^®^ Test*, is based on detecting sialidase activity and was shown to be accurate in non-pregnant women [15,17]. This study sought to validate this rapid sialidase activity detection test for BV diagnosis and evaluate its accuracy as a screening tool for asymptomatic BV in early pregnancy.

## 2. Materials and Methods

### 2.1. Ethics Statement

This study was approved by the Ethics Committee of the Medical University of Vienna (application number: 2115/2019). Our study was conducted following the tenets of the Declaration of Helsinki, Good Scientific Practice guidelines, and the STARD 2015 guidelines for validation studies [19]. All study participants signed an informed consent form before study inclusion. All patient records were pseudo-anonymized and de-identified before analysis.

### 2.2. Setting and Study Population

This prospective validation study was conducted at the Department of Obstetrics and Gynecology, Medical University of Vienna, between 13 February 2020 and 18 March 2021. Our hospital serves about 2800 deliveries per year, including referrals from central and eastern Europe, and is specialized in high-risk pregnancy care. As part of our routine antenatal care, all asymptomatic women registered for a planned delivery at our department undergo routine antenatal infection screening during an early gestation consultation. Asymptomatic pregnant women were considered eligible for study inclusion if they were aged 18–55 years and did not receive antibiotic treatment within the previous two weeks or vaginal medication within the previous 72 h before their presentation at our department. Women with any signs of conspicuous redness, discharge, or vaginal itching were not considered eligible for the study. Demographic data were collected from the obstetric database and patient charts using the PIA Fetal Database, Version 5.6.28.56 (ViewPoint, GE Healthcare, Munich, Germany).

### 2.3. Sampling and Gram Staining Procedure

Vaginal smears were obtained by vaginal fluid collection with sterile cotton-tipped swabs (Puritan 6” Sterile Standard Cotton Swab w/Wooden Handle, Puritan Medical Products, Guilford, ME, USA) from the lateral vaginal wall and posterior fornix vaginae. A vaginal sample was then applied to a microscope slide and Gram-stained. Gram-stained smears were microscopically analyzed by one of five trained and experienced biomedical laboratory assistants specializing in gynecological cytopathology at our laboratory, which is certified according to DIN EN ISO 9001:2008. The protocol involved classification of the vaginal microbiota as described by Nugent et al. [20]. Slides were evaluated under the microscope at up to ×1000 optical magnification. According to this scoring system, a score of 0–3 was regarded as normal, 4–6 as dysbiosis, and 7–10 as BV. Furthermore, the presence of *Candida* species and/or *Trichomonas vaginalis* was microscopically assessed. As part of our routine protocol, participants diagnosed with BV received a 2% clindamycin vaginal cream for six days (primary disease) or 0.3 g oral clindamycin twice daily for seven days (recurrent disease). Participants diagnosed with vulvovaginal candidiasis (VVC; hypha or oidia found in the microscopic evaluation) received 0.1 g clotrimazole vaginal cream for six days, and those with trichomoniasis received 0.5 g metronidazole vaginal cream for seven days [2]. Antibiotic treatment was followed by the vaginal application of *Lactobacillus* spp. (*L. casei rhamnosus*, Lcr35 regenerans) for six days to rebuild the physiological microbiota (e.g., Gynophilus, Mylan Healthcare, Bad Homburg, Germany) [21].

### 2.4. Study Groups

Participants with a Nugent score of 7–10 on Gram-stained screening smears were diagnosed with BV and considered eligible for the study group, whereas those with a Nugent score of 0–3 were considered to be eligible for the control group. Participants of the control group were not allowed to have any evidence of VVC (i.e., oidia or hyphae) or trichomoniasis on microscopic evaluation. Per protocol, 100 participants were included in each group. Following the Gram staining procedure, all participants were tested using the *OSOM^®^ BVBLUE^®^ Test*, as described below.

### 2.5. Point-of-Care Testing Procedure

The *OSOM^®^ BVBLUE^®^ Test* (Sekisui Diagnostics LLC, Burlington, MA, USA) was performed by trained medical staff as recommended by the manufacturer. This test detects elevated sialidase activity in the vaginal fluid. Sialidase is an enzyme produced by bacteria associated with BV [10,11]. For the analysis, a collected vaginal swab was placed in the BV test vessel and gently swirled. The mixture was incubated for 10 min at room temperature (22–24 °C). Two drops of the developer solution were then added, and the swab was gently swirled again. The color reaction was assessed immediately. A green or blue color in the vessel or at the cap of the swab indicated increased sialidase activity, and then the test was considered BV-positive. The test minimum sialidase activity detection limit was 0.25 μg/mL (7.64 U/mL). A yellow color indicated normal sialidase activity without BV, which was considered BV-negative.

### 2.6. Statistical Analysis

Statistical analysis was performed using IBM SPSS Statistics for Windows, Version 27.0 (IBM Corp., Armonk, NY, USA). Graphs and figures were drawn using GraphPad Prism, Version 8.4.1. (GraphPad Software, San Diego, CA, USA) and Lucidchart (Lucid Software Inc., South Jordan, UT, USA). Descriptive statistics summarized the demographic information. Continuous variables are presented as mean ± standard deviation, and ordinally scaled variables are presented as median (interquartile interval) or number (percentage). Binary variables are presented as numbers (percentages). Unpaired, two tailed t-tests were performed for continuous variables, Fisher’s exact tests for binary variables, and Mann–Whitney U-tests for ordinary variables; *p*-values < 0.05 were considered statistically significant. Sensitivity, specificity, positive predictive value (PPV), and negative predictive value (NPV) were calculated for the *OSOM^®^ BVBLUE^®^ Test*. The prevalence of BV was calculated from our screening cohort.

## 3. Results

A total of 1972 pregnant women underwent antenatal infection screening during the study period. Out of these, 1473 women (74.7%) presented with a normal vaginal microbiota (i.e., Nugent score 0–3), 318 women (16.1%) had dysbiosis (i.e., Nugent score 4–6), and 181 women (9.2%) had BV (i.e., Nugent score 7–10). A total of 564 women was excluded from the study according to our protocol. The remaining 100 asymptomatic pregnant women with BV were assigned to the study group, and 100 of 1308 women with a normal microbiota were assigned to the control group after per-protocol inclusion. All women who were asked to participate agreed and were enrolled in the study. Consequently, statistical analysis was performed for a total of 200 study participants. The study inclusion criteria are shown in Figure 1.

The mean maternal age at vaginal sampling was 32.0 ± 5.7 years in the study group, and 33.1 ± 5.1 years in the control group. At that time, the mean gestational age was 17.1 ± 7.0 weeks in the study group, and 15.8 ± 6.2 weeks in the control group. Of the study group, 40 participants (40%) were diagnosed with a Nugent score of 7, and 60 (60%) with a Nugent score of 8. In the control group, 90 participants (90%) were diagnosed with a Nugent score of 0, seven (7%) with a score of 1, one (1%) with a score of 2, and two (2%) with a score of 3. Fourteen participants in the study group (14%) were diagnosed with concomitant *Candida* colonization. Participants in the control group were not allowed to have *Candida* colonization. Participant characteristics are presented in Table 1.

We validated the *OSOM^®^ BVBLUE^®^ Test* against the Gram staining gold standard method and observed that 81/100 (81%) of the study group and 100 (100%) of the control group participants were correctly diagnosed with BV. With a prevalence of 9.2% within our screened cohort, PPV and NPV for the *OSOM^®^ BVBLUE^®^ Test* were 100% and 98.1%, respectively (Table 2). Figure 2 shows the *OSOM^®^ BVBLUE^®^ Test* results according to the Nugent score of the study participants. The number of positive and false-negative tests was not significantly different in women with a Nugent score of 7 and 8 (*p* = 0.60).

## 4. Discussion

Considering the knowledge on the negative effects of BV during pregnancy and the ongoing global burden of PTB, antenatal programs for early BV detection and treatment are highly warranted [7,13]. Point-of-care tests could help, as they are easy to handle and less costly than conventional methods that require laboratory facilities [15,17,18]. Our study evaluated one of these tests in asymptomatic pregnant women by validating it against Gram staining. We demonstrated that the *OSOM^®^ BVBLUE^®^ Test* had high sensitivity, specificity, PPV, and NPV in this context.

It is widely accepted that PTB is a global burden, with prevalence ranging from 5% in European countries to 18% in African countries and a global prevalence of ~11% [22,23]. PTB is the leading cause of neonatal death [3,24] and may entail lifelong effects on neurodevelopmental functions in preterm-born children, such as impaired learning, visual disorders, and an increased risk of cerebral palsy and chronic diseases during adulthood [25]. The economic costs of PTB are immense, considering the need for neonatal intensive care, continued treatment, and educational support [22].

BV during pregnancy was associated with PTB and late miscarriage [1,26,27,28,29]. Treating the abnormal vaginal flora with clindamycin during early pregnancy significantly reduces the risk of PTB and late miscarriage [2,13,30]. Furthermore, the simple health intervention of screening women reduces the risk of adverse pregnancy outcomes, even when they are asymptomatic [7,13]. However, from a clinical perspective, antenatal care is usually performed in outpatient offices, where modern diagnostic procedures are expensive, time-consuming, and associated with a time latency before the woman receives her test results [18,31]. When the results arrive, it might already be too late to avert a flourishing infection of the lower genital tract.

One approach to overcome this issue might be the use of point-of-care tests. These rapid detection assays are available on the market to detect several pathogens, including BV, VVC, and trichomoniasis. In fact, these tests represent a quick and accessible alternative to the conventional diagnostic methods [15,18,31,32,33]. Furthermore, point-of-care tests are cheaper than culture methods or Gram staining when considering the costs of medical staff and facilities [33]. These tests can also help avoid unnecessary patient consultations, diagnostic procedures, and treatments, thereby reducing healthcare costs [17,18,33,34]. On a practical level, point-of-care tests present an opportunity for healthcare professionals to screen women for vaginal infections without the involvement of hospitals and laboratory facilities. Women can also perform the test by themselves. Such a self-screening approach has already been described in Germany, where pregnant women self-measured pH by a testing glove. This self-screening approach was also shown to significantly reduce the rate of PTB [35]. However, an elevated pH of ≥4.7 is relatively unspecific and might not be due to BV. This underlines the need for more precise screening tools.

For this study, we decided to use the *OSOM^®^ BVBLUE^®^ Test*, as it has already been validated for BV diagnosis in symptomatic, non-pregnant women [15,17,36]. For these women, the available literature suggested a sensitivity, specificity, PPV, and NPV of 88.0–97.6%, 95.0–97.8%, 91.7–98.4%, and 91.0–97.6%, respectively [15,17,36]. Amsel criteria were inferior to the test in detecting BV in two of the three studies that have evaluated this test [15,36]. For this reason, we chose not to compare the test to the Amsel criteria, but to validate it against Gram staining, which is the gold standard method during antenatal care at our department.

Our study is distinctly different from previously published studies as we evaluated this test for screening asymptomatic pregnant women. The PPV and NPV that we present herein are important measures to demonstrate the effect of this public health intervention [37,38]. The PPV of 100% and NPV of 98.1% that we found suggest that the *OSOM^®^ BVBLUE^®^ Test* is highly accurate in screening asymptomatic pregnant women for BV.

We are aware that our study and point-of-care tests have several limitations. The predominant limitation of the test is the interpretation of the test results in the vessel. The yellow color, a negative result indicator, was difficult to distinguish from the green color, a BV indicator. Hence, our interpretation may have been incorrect, which, however, would only have affected four cases in our study. Moreover, we did not include culture methods, which might be considered a limitation; we chose this procedure for cost reduction, aiming to compare the test to Gram staining as the gold standard method. Culture methods would reproduce the vaginal ecosystem diversity [16], but diagnosis would be less easy to interpret. Moreover, *Gardnerella vaginalis* is part of the physiological lactobacilli-dominated microbiota of healthy women, albeit only in small numbers, and therefore detection of *Gardnerella vaginalis* in culture methods would not necessarily indicate BV [39]. On the other hand, microscopy of Gram-stained smears is an operator-dependent procedure, which might have also influenced our results.

The strengths of our study include the relatively large number of screened women and homogeneous pregnancy care and study settings. It could have been interesting to know how many of the screened women experienced PTB. However, this was not an outcome of this study, as all women with BV received adequate treatment, and the available literature in this regard is very clear.

## 5. Conclusions

We demonstrated that sialidase activity detection by the *OSOM^®^ BVBLUE^®^ Test* was an accurate method for BV diagnosis in asymptomatic pregnant women compared to Gram staining. Our results support using this point-of-care test as a reliable and easy-to-use screening tool for BV during antenatal care. Together with other point-of-care tests for potentially harmful pathogens, this test enables early screening, detection, and treatment of BV during pregnancy and might therefore contribute to a future reduction in the rate of preterm birth.

## Figures and Tables

**Figure 1 jcm-10-02275-f001:**
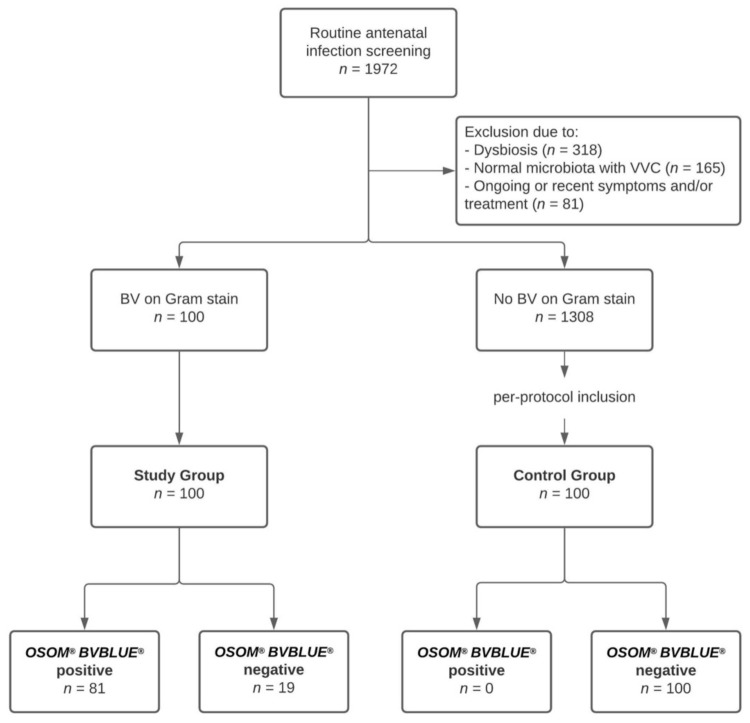
Inclusion criteria of 200 asymptomatic pregnant women, screened for BV using Gram-stained smears and the *OSOM^®^ BVBLUE^®^ Test* between 13 February 2020 and 18 March 2021. BV, bacterial vaginosis. VVC, vulvovaginal candidosis.

**Figure 2 jcm-10-02275-f002:**
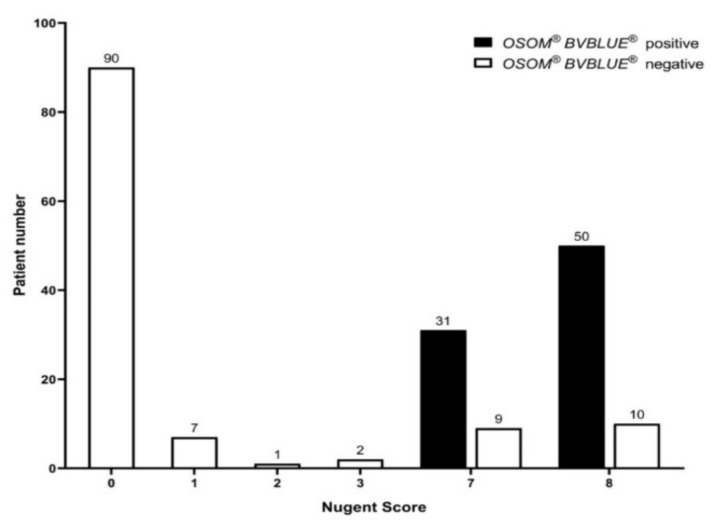
Results of the *OSOM^®^ BVBLUE^®^ Test* according to the Nugent score of 200 asymptomatic pregnant women.

**Table 1 jcm-10-02275-t001:** Characteristics of 200 asymptomatic pregnant women, screened for BV using Gram-stained smears and the *OSOM^®^ BVBLUE^®^ Test* (data presented as numbers and percentages, mean ± standard deviation, or median (range); unpaired two tailed t-tests performed for continuous variables, Fisher’s exact tests for binary variables, and Mann–Whitney U-Tests for ordinary variables; BV, bacterial vaginosis; PTB, preterm birth).

Characteristic	Study Group (*n* = 100)	Control Group (*n* = 100)	All (*n* = 200)	*p*-Value
Participant Age (years)	32.0 ± 5.7	33.1 ± 5.1	32.6 ± 5.4	0.16
Gravidity	2 (1–11)	2 (1–7)	2 (1–11)	0.36
Parity	1 (0–8)	1 (0–6)	1 (0–8)	0.22
Smoking				0.05
Yes	21 (21.0%)	10 (10.0%)	31 (15.5%)	
No	79 (79.0%)	90 (90.0%)	169 (84.5%)	
PTB in the previous pregnancy				1.00
Yes	6 (6.0%)	5 (5.0%)	11 (5.5%)	
No	94 (94.0%)	95 (95.0%)	189 (94.5%)	
Gestational weeks at sampling	17.1 ± 7.0	15.8 ± 6.2	16.4 ± 6.7	0.17
*Candida* colonization				<0.001
Yes	14 (14.0%)	0 (0.0%)	14 (7.0%)	
No	86 (86.0%)	100 (100.0%)	186 (93.0%)	

**Table 2 jcm-10-02275-t002:** Validation of the *OSOM^®^ BVBLUE^®^ Test* in comparison to Gram staining in 200 asymptomatic pregnant women screened for BV.

Parameter	Study Group (*n* = 100)	Control Group (*n* = 100)	Total (*n* = 200)
Positive test	81	0	81
Negative test	19	100	119
Total	100	100	200

Sensitivity = 81.0%, specificity = 100.0%, positive predictive value = 100.0%, negative predictive value = 98.1%; BV prevalence: 9.2%. BV, bacterial vaginosis.

## Data Availability

The data are available on request to the corresponding author.

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
