# Peer review of "Screening Pregnant Women for Bacterial Vaginosis Using a Point-of-Care Test: A Prospective Validation Study"

_jcm, 2021, doi:10.3390/jcm10112275_

Round 1

Reviewer 1 Report

The manuscript by Philipp Foessleitner et al. describes interesting data about a POCT for diagnosis of BV among pregnant women.

This manuscript suffers from a lack of information and the manuscript deserves revision before possible resubmission.

Global :

Italicize: "et al."; "i.e."

Prefer passive form

Methods : 

How was determined the number of included patients ? Please justify.

How were considered the risk of false-positive/negative results for Gram staining reading? Moreover, it seems that a medical microbiologist deserves to be including in the manuscript. It is an important limitation of the manuscript that both the methods are operator-dependant, limiting its extrapolation.

Have the operator been aware of the results of Gram-staining examination before OSOM BVBlue testing ? If not explain how was maintained the blindness.

Line 104 : give precision about the Lactobacillus? Is it asingle strain? a cocktail ? What is the manufacturer?

Line 117 : Is a sentence lacking "The"?

Statistical analysis :  PLease indicate all the version of softwares used for the study. Moreover, due to the case-control study design of the study, it cannot be considered PPV and NPV, even if the authors use an estimation of the prevalence. Limit your conclusion to likelihood ratio (and it will ease your comparison with the litterature results).

Results : 

Was a control considered for patient with a Nugent's score of 3 to 7?

How were considered patients included once in the study? Have they been excluded? Precise.

No patient present a score of 9 or 10?

For patient with descrepant results (n=19) precise the repartition of their Nugent score. Have the authors consider to do a third method.

Discussion : 

It is not clear how the authors consider the benefit to use this method compared to the Nugent score. The latter is rapid, not expensive and bring sufficient information in a timely maneer. I do not think that this test could overcome these characteristics, especially with that much discrepant results.

Author Response

REVIEWER 1

Reviewer 1, Comment 1:

The manuscript by Philipp Foessleitner et al. describes interesting data about a POCT for diagnosis of BV among pregnant women. This manuscript suffers from a lack of information and the manuscript deserves revision before possible resubmission.

Global: Italicize: "et al."; "i.e."

Response to Reviewer 1, Comment 1:

We appreciate the helpful comments of Reviewer 1 that have been addressed accordingly in the revised version of our manuscript.

"et al." and "i.e.," have been italicized throughout the manuscript.

Reviewer 1, Comment 2:

Methods: How was determined the number of included patients? Please justify.

Response to Reviewer 1, Comment 2:

We thank Reviewer 1 for the comment. The number of included study participants was justified considering (i) the patient volume at our department, (ii) the prevalence of BV within our patient collective, (iii) the planned recruitment period, as well as (iv) the number of participants in the available literature for non-pregnant women [1,2].

Reviewer 1, Comment 3:

How were considered the risk of false-positive/negative results for Gram staining reading? Moreover, it seems that a medical microbiologist deserves to be including in the manuscript. It is an important limitation of the manuscript that both the methods are operator-dependent, limiting its extrapolation.

Response to Reviewer 1, Comment 3:

Thank you for this comment. As described in the Methods section of our manuscript, the aim of our study was to validate the point-of-care test OSOM® BVBLUE® Test and evaluate its accuracy as a screening tool for asymptomatic BV in early pregnancy. The test was thereby validated against Gram stain, which is widely considered the gold standard method for the diagnosis of BV [3]. For this reason, we did not separately consider the risk of false-positive or false-negative results on Gram stain. As described in the Limitations section of our manuscript, it would have been beneficial to include culture methods, which we, however, did not do for the sake of cost reduction, and as culture methods do not allow the clear identification of pathogenic strains among the bacterial variety in the vaginal ecosystem. As suggested by the Reviewer, we have now added a sentence in the Limitations section of our manuscript, stating that microscopy on Gram-stained smears, as we used it, is an operator-dependent procedure. We believe that this should now be clear for the readership (page 7, lines 245–246).

There was no medical microbiologist included in the study, as we did not perform cultural methods. Therefore it was not justified to include a microbiologist, who was not involved in the study, as an author. As described in our manuscript (page 2, lines 78–80), Gram-staining was performed as part of our clinical routine. The point-of-care tests were performed by trained and certified medical staff at our department, who were all listed as authors (page 3, lines 119–120). However, we have now acknowledged the work our laboratory assistants within the Acknowledgment section of our manuscript.

Reviewer 1, Comment 4:

Have the operator been aware of the results of Gram-staining examination before OSOM BVBlue testing? If not explain how was maintained the blindness.

Response to Reviewer 1, Comment 4:

Yes, indeed we were aware of the results of Gram-staining examination before performing the point-of-care test. As described in the Methods section of our manuscript, microscopic evaluation of Gram-stained smears was performed first, followed by the OSOM® BVBLUE® Test, but only in case of study eligibility: i.e., Nugent score of 0–3 without any sign of VVC for the control group, or a Nugent score of 7–10 for the study group (page 3, lines 111–113).

Reviewer 1, Comment 5:

Line 104: give precision about the Lactobacillus? Is it a single strain? a cocktail? What is the manufacturer?

Response to Reviewer 1, Comment 5:

We thank Reviewer 1 for this interesting question. We prescribe a formulation of vaginally applied capsules containing at least 109 colony‐forming units of living L. casei rhamnosus (Lcr35 regenerans) each (e.g., Gynophilus, Mylan Healthcare GmbH, Bad Homburg, Germany). We added this information to the Methods section of the revised version of our manuscript (page 3, lines 106–108).

Reviewer 1, Comment 6:

Line 117: Is a sentence lacking "The"?

Response to Reviewer 1, Comment 6:

We apologize for this mistake. The sentence was revised accordingly.

Reviewer 1, Comment 7:

Statistical analysis:  Please indicate all the version of softwares used for the study. Moreover, due to the case-control study design of the study, it cannot be considered PPV and NPV, even if the authors use an estimation of the prevalence. Limit your conclusion to likelihood ratio (and it will ease your comparison with the literature results).

Response to Reviewer 1, Comment 7:

Thank you for your question. Statistical analysis was performed using IBM SPSS Statistics for Windows, Version 27.0 (IBM Corp., Armonk, NY, USA). Graphs and figures were drawn using GraphPad Prism, Version 8.4.1. (GraphPad Software, San Diego, CA, USA). We have added the information about the version to our revised manuscript (page 3, lines 132–135).

After consulting our statistician, we want to point out that case control studies can be used to calculate the PPV and NPV. However, since these studies usually oversample subjects with disease, and as the relative proportion of cases and controls does not reflect the prevalence of the disease in the target population, disease prevalence should be estimated from a separate study [4]. We decided to use the prevalence reported in the cross-sectional study from Farr et al. [5], because subjects were drawn from a comparable population at the same institution. The calculation of PPV and NPV in our study meets the statistical criteria needed for validating one method with another, which we did comparing the point-of-care test with Gram stain. We apologize if this remained unclear, which is why we also corrected the title of our manuscript to: “Screening pregnant women for bacterial vaginosis using a point-of-care test: a prospective validation study”. We believe that this should now be clear for the readership.

Reviewer 1, Comment 8:

Results: Was a control considered for patient with a Nugent's score of 3 to 7?

Response to Reviewer 1, Comment 8:

Thank you for your question. As stated in Response to Reviewer 1, Comment 9, only pregnant women with BV diagnosed on the Gram stain (Nugent score of 7–10) or with a Gram staining revealing a normal vaginal microbiota (Nugent score of 0–3, no signs of VVC) were considered to be eligible for study inclusion. Indeed, we recommend a control vaginal smear after 4–6 weeks to all women.

Reviewer 1, Comment 9:

How were considered patients included once in the study? Have they been excluded? Precise.

Response to Reviewer 1, Comment 9:

Thank you for this important comment. We apologize if our description of the recruitment procedure might have been misleading. To clarify: vaginal smears of asymptomatic pregnant women undergoing routine antenatal screening at our department and fulfilling the inclusion criteria were routinely evaluated using Gram stain. In case of BV on Gram stain (Nugent score 7–10) or in case of a normal vaginal microbiota (Nugent score 0–3 without signs of VVC), patients were considered to be eligible and asked to participate in the study. Following informed consent, we performed the OSOM®BVBLUE® Test. To be as specific as possible, we have therefore changed “cross-sectional” in the title of our manuscript, which therefore might have been misleading. In addition, we have revised the Methods section to describe the recruitment procedure in more detail as follows: “Participants with a Nugent score of 7–10 on Gram-stained screening smears were diagnosed with BV and considered eligible for the study group, whereas those with a Nugent score of 0–3 were considered to be eligible for the control group.” (page 3, lines 111–113). We believe that this should now be clear for the readership and we apologize if this point remained unclear before.

Reviewer 1, Comment 10:

No patient present a score of 9 or 10?

Response to Reviewer 1, Comment 10:

Indeed, there was no women presenting with a Nugent score of 9–10 during our study period without the presence of Candida on Gram-stained smears. However, we consider this reasonable, as our study participants were all asymptomatic, which makes it less likely to find a high Nugent score, as these women would have been more likely to suffer from any symptoms.

Reviewer 1, Comment 11:

For patient with discrepant results (n=19) precise the repartition of their Nugent score. Have the authors consider to do a third method.

Response to Reviewer 1, Comment 11:

Thank you for your question. Of the 19 participants in the study group with false-negative results, 9 women (47%) were diagnosed with a Nugent score of 7, and 10 women (53%) had a Nugent score of 8 on Gram stain. As suggested by Reviewer 2, we have calculated a Fisher’s exact test, which revealed no statistically significant difference regarding the number of false-negative test results in patients with Nugent score 7 and 8. We have now added this information to the Results section of the revised version of the manuscript (page 5, lines 172–173). Please also see our Response to Reviewer 2, Comment 11.

Reviewer 1, Comment 12:

Discussion: It is not clear how the authors consider the benefit to use this method compared to the Nugent score. The latter is rapid, not expensive and bring sufficient information in a timely maneer. I do not think that this test could overcome these characteristics, especially with that much discrepant results.

Response to Reviewer 1, Comment 12:

Thank you for your comment, which is discussed as a crucial point in our manuscript. Microscopic evaluation of Gram-stained smears is widely considered the gold standard method for BV diagnosis [3]. However, Gram staining requires a laboratory and trained and experienced staff, which is not available in all medical facilities, and especially not in outpatients offices. Furthermore, these facilities and specialized staff are both cost intensive. Point-of-care tests, on the contrary, are cheap and easy to perform, and they do not require specialized facilities, providing results within 10 minutes [1,2,6]. This could be of interest for outpatient offices or self-testing of women, as already pointed out in the Discussion section of our manuscript [6,7]. In our study, we were able demonstrate that the OSOM® BVBLUE® Test showed high concordance with Gram stain (page 7, lines 253–259).

--

REFERENCES

  1. Myziuk, L.; Romanowski, B.; Johnson, S.C. BVBlue test for diagnosis of bacterial vaginosis. J Clin Microbiol 2003, 41, 1925-1928, doi:10.1128/jcm.41.5.1925-1928.2003.
  2. Bradshaw, C.S.; Morton, A.N.; Garland, S.M.; Horvath, L.B.; Kuzevska, I.; Fairley, C.K. Evaluation of a point-of-care test, BVBlue, and clinical and laboratory criteria for diagnosis of bacterial vaginosis. J Clin Microbiol 2005, 43, 1304-1308, doi:10.1128/jcm.43.3.1304-1308.2005.
  3. Vardar, E.; Maral, I.; Inal, M.; Ozgüder, O.; Tasli, F.; Postaci, H. Comparison of Gram stain and Pap smear procedures in the diagnosis of bacterial vaginosis. Infect Dis Obstet Gynecol 2002, 10, 203-207, doi:10.1155/s1064744902000236.
  4. Ying, G.S.; Maguire, M.G.; Glynn, R.J.; Rosner, B. Calculating Sensitivity, Specificity, and Predictive Values for Correlated Eye Data. Invest Ophthalmol Vis Sci 2020, 61, 29, doi:10.1167/iovs.61.11.29.
  5. Farr, A.; Kiss, H.; Hagmann, M.; Marschalek, J.; Husslein, P.; Petricevic, L. Routine Use of an Antenatal Infection Screen-and-Treat Program to Prevent Preterm Birth: Long-Term Experience at a Tertiary Referral Center. Birth (Berkeley, Calif.) 2015, 42, 173-180, doi:10.1111/birt.12154.
  6. Foessleitner, P.; Kiss, H.; Deinsberger, J.; Ott, J.; Zierhut, L.; Farr, A. Validation of the SavvyCheck™ Vaginal Yeast Test for Screening Pregnant Women for Vulvovaginal Candidosis: A Prospective, Cross-Sectional Study. J Fungi (Basel) 2021, 7, doi:10.3390/jof7030233.
  7. Hoyme, U.B.; Saling, E. Efficient prematurity prevention is possible by pH-self measurement and immediate therapy of threatening ascending infection. Eur J Obstet Gynecol Reprod Biol 2004, 115, 148-153, doi:10.1016/j.ejogrb.2004.02.038.

Reviewer 2 Report

Foessleitner et al present a validation study of a sialidase activity detection kit (OSOM BVBLUE) as a screening tool for bacterial vaginosis in pregnant women. They validate the easy-to-use screening tool against the gold standard laboratory-dependent Gram-staining procedure. The authors present a balanced study population with 200 participants, of which 100 had a Nugent score of 0-3 (Control) and 100 had a Nugent score of 7-10 (Study).

Foessleitner et al raise some very valid points in the manuscript and it is my opinion that the manuscript should be reconsidered for publication after major revisions.

Major concerns:

I am concerned about the recruitment process presented in the study and there are several points that I would like to have clarified in relation to this:

The Authors state that 200 asymptomatic women were enrolled in the study and none of them were excluded, while also assuming that 7.4% of the women will have BV (the reference used reports 8.7%). Why do they not report a measured prevalence of BV in this study? And if 7.4% prevalence is reasonable, more than 1300 women would need to be enrolled to reach 100 women with BV. As I understand the study design the vaginal smear and the OSOM BVBLUE screen were performed at the same visit, so how did the authors manage to have two equally sized groups? If participants have had the OSOM BVBLUE screen and then been excluded what was the reason for exclusion?

I understand that as all participants were asymptomatic, it is reasonable that none had a Nugent score above 8, but if no one were excluded from the study why did no participant have a Nugent score of 4-6? I think that it is very reasonable to exclude participants with an intermediate score as they might introduce a significant amount of noise, but this has to be reported.

It should also be reported how many were invited to participate, how many were eligible/excluded based on the criteria in section 2.2, and how many were excluded based on their Nugent score.

The authors present the characteristics of the two groups in table 1, but does not present any statistical comparison of the characteristics. It is important that this is added.

Why does the authors need to assume a BV prevalence in order to calculate PPV and NPV? These should be based on the participants in the study, which means that PPV is correct (100%) the NPV is only 84% in this study and not 98.5% (line 220).

In the discussion, line 218-219, the authors states that the PPV and NPV are the most important statistical measures as the goal is to minimise the risk of PTB. I do not agree with this statement, as a healthcare perspective a point-of-care test could be used as an initial screen to guide whether a laboratory test should be performed. Therefore, some false positives would be acceptable, but a 19% false negative rate does seem high to me.

Minor concerns:

Line 36: Reference 6 states that 25-40% of PTM is due to infections, not specifically vaginal infections or BV, and in reference 12 less than 30% of mothers that had PTM had an abnormal flora. I think the point is relevant so I suggest changing the statement to talk about the increased risk PTM when having BV and use different references (https://pubmed.ncbi.nlm.nih.gov/7491137/ or something more recent)  

Line 43: change to “…, and to modify …”

Line 113: room temperature listed as 17-37°C. Is this a mistake or were your lab very warm? If the temperature is correct, just remove then mention of room temperature.

Line 117: The paragraph ends with a hanging “The”

Line 160: Please include a statistical test to show that there is no statistical difference between the number of false negatives between Nugent score 7 and 8 (The Fisher exact test statistic value is 0.6037, so not significant)

Figure 2: The bar at Nugent score 2 looks black. I suggest that the figure is updated to avoid any confusion.

Author Response

REVIEWER 2

Reviewer 2, Comment 1:

Foessleitner et al present a validation study of a sialidase activity detection kit (OSOM BVBLUE) as a screening tool for bacterial vaginosis in pregnant women. They validate the easy-to-use screening tool against the gold standard laboratory-dependent Gram-staining procedure. The authors present a balanced study population with 200 participants, of which 100 had a Nugent score of 0-3 (Control) and 100 had a Nugent score of 7-10 (Study).

Foessleitner et al raise some very valid points in the manuscript and it is my opinion that the manuscript should be reconsidered for publication after major revisions.

Major concerns: I am concerned about the recruitment process presented in the study and there are several points that I would like to have clarified in relation to this:

The Authors state that 200 asymptomatic women were enrolled in the study and none of them were excluded, while also assuming that 7.4% of the women will have BV (the reference used reports 8.7%). Why do they not report a measured prevalence of BV in this study? And if 7.4% prevalence is reasonable, more than 1300 women would need to be enrolled to reach 100 women with BV. As I understand the study design the vaginal smear and the OSOM BVBLUE screen were performed at the same visit, so how did the authors manage to have two equally sized groups? If participants have had the OSOM BVBLUE screen and then been excluded what was the reason for exclusion?

Response to Reviewer 2, Comment 1:

We thank Reviewer 2 for the valuable feedback and improvement suggestions. We apologize if the description of our study procedure might have been misleading. We have revised the Methods section as suggested, and we hope that the recruitment procedure and study design should now be clear for the readers. Please see our Response to Reviewer 1, Comment 9.

Regarding the prevalence of BV in our study cohort: Farr et al. reported a prevalence of 7.4% in the overall pregnant cohort in their study, which was also conducted at our institution. The prevalence of 8.7% that was suggested by Reviewer 2 represents the BV prevalence in the preterm birth group only [5]. We decided to use the prevalence reported in the cross-sectional study of Farr et al., as cases were drawn from a comparable population. We cannot describe the total number of women with BV in our collective, as this was not part of our study. We did not perform a study evaluating the microbiota of pregnant women at our institution (as Farr et al. did in their study), but we aimed to validate a point-of-care test with Gram stain in women with and without BV on Gram-stained smears. We hope this point is now clear and we apologize if it might have been misleading.

We also want to point out that swabs for Gram staining and the point-of-care test were taken at the same visit. However, we performed Gram stain first and, if women were suitable for enrolment in either the study or the control group, the OSOM® BVBLUE® Test was subsequently conducted. The arising waiting time for the evaluation of the Gram stain was spent within the routine antenatal care and booking for delivery at our institution. Please also see Response to Reviewer 1, Comment 11.

Reviewer 2, Comment 2:

I understand that as all participants were asymptomatic, it is reasonable that none had a Nugent score above 8, but if no one were excluded from the study why did no participant have a Nugent score of 4-6? I think that it is very reasonable to exclude participants with an intermediate score as they might introduce a significant amount of noise, but this has to be reported.

Response to Reviewer 2, Comment 2:

Thank you for this important question. As stated in our Response to Reviewer 1, Comment 8 and our Response to Reviewer 1, Comment 9, there was no study participant with a Nugent score of 4–6 included.

Reviewer 2, Comment 3:

It should also be reported how many were invited to participate, how many were eligible/excluded based on the criteria in section 2.2, and how many were excluded based on their Nugent score.

Response to Reviewer 2, Comment 3:

Please see our Response to Reviewer 1, Comment 9. For the sake of method validation, we asked 200 asymptomatic pregnant women to participate in our study. All women who were asked to participate agreed, and none of them were excluded; we have now included this information in the revised version of our manuscript (page 4, lines 145–146). Regarding the residual numbers, we can only estimate those of the non-eligible women (i.e., due to candidosis and/or intermediate microbiota). However, this was not of interest in our study, and has already been published before [5].

Reviewer 2, Comment 4:

The authors present the characteristics of the two groups in table 1, but does not present any statistical comparison of the characteristics. It is important that this is added.

Response to Reviewer 2, Comment 4:

We thank Reviewer 2 for raising this point. We have added a column with the respective p-values to Table 1. Please find the adapted Table 1 below and in our revised manuscript.

Characteristic

Study group

(n = 100)

Control group

 (n = 100)

All

(n = 200)

p-value

Participant Age (years)

32.0 ± 5.7

33.1 ± 5.1

32.6 ± 5.4

0.16

Gravidity

2 [1–11]

2 [1–7]

2 [1–11]

0.36

Parity

1 [0–8]

1 [0–6]

1 [0–8]

0.22

Smoking

Yes

No

21 (21.0%)

79 (79.0%)

10 (10.0%)

90 (90.0%)

31 (15.5%)

169 (84.5%)

0.05

PTB in the previous pregnancy

Yes

No

6 (6.0%)

94 (94.0%)

5 (5.0%)

95 (95.0%)

11 (5.5%)

189 (94.5%)

1.00

Gestational weeks at sampling

17.1 ± 7.0

15.8 ± 6.2

16.4 ± 6.7

0.17

Candida colonization

Yes

No

14 (14.0%)

86 (86.0%)

0 (0.0%)

100 (100.0%)

14 (7.0%)

186 (93.0%)

<0.001

Reviewer 2, Comment 5:

Why does the authors need to assume a BV prevalence in order to calculate PPV and NPV? These should be based on the participants in the study, which means that PPV is correct (100%) the NPV is only 84% in this study and not 98.5% (line 220).

Response to Reviewer 2, Comment 5:

The study population was predefined to consist of an equal amount of BV positive and negative patients. Hence, the relative proportion of cases and controls does not reflect the prevalence of the disease in the target population and it would therefore not be appropriate to calculate the PPV and NPV directly [4]. Instead, it is a common technical procedure to estimate the disease prevalence of a target population from a separate study [4]. Since subjects were drawn from a population similar to the cross-sectional study by Farr et al. [5], we decided to use the therein reported prevalence. The use of NPV and PPV in this context, validating a method with another, is a common procedure according to our statistician. Please also see Response to Reviewer 1, Comment 7.

Reviewer 2, Comment 6:

In the discussion, line 218-219, the authors states that the PPV and NPV are the most important statistical measures as the goal is to minimize the risk of PTB. I do not agree with this statement, as a healthcare perspective a point-of-care test could be used as an initial screen to guide whether a laboratory test should be performed. Therefore, some false positives would be acceptable, but a 19% false negative rate does seem high to me.

Response to Reviewer 2, Comment 6:

We thank Reviewer 2 for raising this point. We did not intent to suggest that sensitivity and specificity do not represent important measures. We therefore adapted the paragraph to avoid confusion of the reader. We agree with Reviewer 2 that a sensitivity of 81% and therefore a false negative rate of 19% is not optimal. However, considering the prevalence of BV in our study was artificially set at 50%, but the prevalence in the general population is only 7.4%, and the resulting NPV is 98.5%, indicating a false negative rate of only 1.5% within the general population, the test has shown paramount efficiency when being used as a screening test. We have adapted paragraph as follows: “The PPV and NPV that we present herein are important measures to demonstrate the effect of this public health intervention [8,9]. The PPV of 100% and NPV of 98.5% that we found, both suggest that the OSOM® BVBLUE® Test is highly accurate in screening asymptomatic pregnant women for BV.” (page 6, lines 230–233).

Reviewer 2, Comment 7:

Minor concerns: Line 36: Reference 6 states that 25-40% of PTM is due to infections, not specifically vaginal infections or BV, and in reference 12 less than 30% of mothers that had PTM had an abnormal flora. I think the point is relevant so I suggest changing the statement to talk about the increased risk PTM when having BV and use different references (https://pubmed.ncbi.nlm.nih.gov/7491137/ or something more recent) 

Response to Reviewer 2, Comment 7:

We thank Reviewer 2 for this valuable suggestion. We have revised the paragraph and updated the references in the revised form of our manuscript as suggested: “In fact, maternal infections during pregnancy account for up to 40% of PTB cases [10]. Vaginal dysbiosis and consecutive BV increase the risk of PTB, which is of particular interest since its incidence is relatively high among women of reproductive age [5,11].” (page 1, lines 36–39).

Reviewer 2, Comment 8:

Line 43: change to “…, and to modify …”

Response to Reviewer 2, Comment 8:

We thank Reviewer 2 for this suggestion and have revised the sentence as follows: “Sialic acid is used by these bacteria to adhere to cellular and inert surfaces as sources of nutrition, and to modify the immune response and normal mucus barrier.” (page 1–2, lines 43–45).

Reviewer 2, Comment 9:

Line 113: room temperature listed as 17-37°C. Is this a mistake or were your lab very warm? If the temperature is correct, just remove then mention of room temperature.

Response to Reviewer 2, Comment 9:

We thank Reviewer 2 for making as aware of this mistake. We now report the actual temperature within our laboratory, which is 22–24°C and kept stabile under continuous climate control (page 3, lines 123–124).

Reviewer 2, Comment 10:

Line 117: The paragraph ends with a hanging “The”

Response to Reviewer 2, Comment 10:

The hanging “The” has been removed.

Reviewer 2, Comment 11:

Line 160: Please include a statistical test to show that there is no statistical difference between the number of false negatives between Nugent score 7 and 8 (The Fisher exact test statistic value is 0.6037, so not significant)

Response to Reviewer 2, Comment 11:

We have performed a Fisher’s exact test and can confirm p=0.6037. Therefore, the difference is not statistically significant. We have included this information in the revised manuscript (page 5, lines 172–173).

Reviewer 2, Comment 12:

Figure 2: The bar at Nugent score 2 looks black. I suggest that the figure is updated to avoid any confusion.

Response to Reviewer 2, Comment 12:

We thank Reviewer 2 for this suggestion, and we apologize if this might have caused confusion. We therefore updated Figure 2 accordingly in our revised manuscript. 

--

REFERENCES

  1. Myziuk, L.; Romanowski, B.; Johnson, S.C. BVBlue test for diagnosis of bacterial vaginosis. J Clin Microbiol 2003, 41, 1925-1928, doi:10.1128/jcm.41.5.1925-1928.2003.
  2. Bradshaw, C.S.; Morton, A.N.; Garland, S.M.; Horvath, L.B.; Kuzevska, I.; Fairley, C.K. Evaluation of a point-of-care test, BVBlue, and clinical and laboratory criteria for diagnosis of bacterial vaginosis. J Clin Microbiol 2005, 43, 1304-1308, doi:10.1128/jcm.43.3.1304-1308.2005.
  3. Vardar, E.; Maral, I.; Inal, M.; Ozgüder, O.; Tasli, F.; Postaci, H. Comparison of Gram stain and Pap smear procedures in the diagnosis of bacterial vaginosis. Infect Dis Obstet Gynecol 2002, 10, 203-207, doi:10.1155/s1064744902000236.
  4. Ying, G.S.; Maguire, M.G.; Glynn, R.J.; Rosner, B. Calculating Sensitivity, Specificity, and Predictive Values for Correlated Eye Data. Invest Ophthalmol Vis Sci 2020, 61, 29, doi:10.1167/iovs.61.11.29.
  5. Farr, A.; Kiss, H.; Hagmann, M.; Marschalek, J.; Husslein, P.; Petricevic, L. Routine Use of an Antenatal Infection Screen-and-Treat Program to Prevent Preterm Birth: Long-Term Experience at a Tertiary Referral Center. Birth (Berkeley, Calif.) 2015, 42, 173-180, doi:10.1111/birt.12154.
  6. Foessleitner, P.; Kiss, H.; Deinsberger, J.; Ott, J.; Zierhut, L.; Farr, A. Validation of the SavvyCheck™ Vaginal Yeast Test for Screening Pregnant Women for Vulvovaginal Candidosis: A Prospective, Cross-Sectional Study. J Fungi (Basel) 2021, 7, doi:10.3390/jof7030233.
  7. Hoyme, U.B.; Saling, E. Efficient prematurity prevention is possible by pH-self measurement and immediate therapy of threatening ascending infection. Eur J Obstet Gynecol Reprod Biol 2004, 115, 148-153, doi:10.1016/j.ejogrb.2004.02.038.
  8. Akobeng, A.K. Understanding diagnostic tests 1: sensitivity, specificity and predictive values. Acta Paediatr 2007, 96, 338-341, doi:10.1111/j.1651-2227.2006.00180.x.
  9. Trevethan, R. Sensitivity, Specificity, and Predictive Values: Foundations, Pliabilities, and Pitfalls in Research and Practice. Front Public Health 2017, 5, 307, doi:10.3389/fpubh.2017.00307.
  10. Romero, R.; Espinoza, J.; Kusanovic, J.P.; Gotsch, F.; Hassan, S.; Erez, O.; Chaiworapongsa, T.; Mazor, M. The preterm parturition syndrome. Bjog 2006, 113 Suppl 3, 17-42, doi:10.1111/j.1471-0528.2006.01120.x.
  11. Lamont, R.F.; Nhan-Chang, C.L.; Sobel, J.D.; Workowski, K.; Conde-Agudelo, A.; Romero, R. Treatment of abnormal vaginal flora in early pregnancy with clindamycin for the prevention of spontaneous preterm birth: a systematic review and metaanalysis. Am J Obstet Gynecol 2011, 205, 177-190, doi:10.1016/j.ajog.2011.03.047.

Reviewer 3 Report

Title: Screening of Pregnant Women for Bacterial Vaginosis Using a Point-of-Care Test: A Prospective, Cross-Sectional Study.

General comments: I think manuscript is well written and worthy of publication. Authors in this study evaluated the accuracy of a point-of-care OSOM BVBLUE, sialidase activity detection test for asymptomatic pregnant women compared to gram stain. They enrolled 200 pregnant participants with 100 confirmed BV positive and 100 healthy controls with gram stain and observed a sensitivity and specificity of 81% and 100%, respectively. I support the publication this manuscript given the strength of data and consistency with other literature despite minor revisions.

Major Critique-

  1. As mentioned in the limitations of this test “yellow color, a negative result indicator, was difficult to distinguish from the green color, a BV indicator. Hence, our interpretation may have been incorrect in some cases.” How many cases your interpretation of the results could be incorrect? Does it change the outcome of your study? It will be beneficial to mention the rate of incorrect interpretation.

Minor Critique-

  1. Line 216-217, “this study was distinctly different from previously published study”. Please explain the difference clearly and provide the reference for those studies.

Author Response

REVIEWER 3

Reviewer 3, Comment 1:

I think manuscript is well written and worthy of publication. Authors in this study evaluated the accuracy of a point-of-care OSOM BVBLUE, sialidase activity detection test for asymptomatic pregnant women compared to gram stain. They enrolled 200 pregnant participants with 100 confirmed BV positive and 100 healthy controls with gram stain and observed a sensitivity and specificity of 81% and 100%, respectively. I support the publication this manuscript given the strength of data and consistency with other literature despite minor revisions.

Major Critique: As mentioned in the limitations of this test “yellow color, a negative result indicator, was difficult to distinguish from the green color, a BV indicator. Hence, our interpretation may have been incorrect in some cases.” How many cases your interpretation of the results could be incorrect? Does it change the outcome of your study? It will be beneficial to mention the rate of incorrect interpretation.

Response to Reviewer 3, Comment 1:

We thank Reviewer 3 for the valuable comments and suggestions. Indeed, the mentioned difficulty to distinguish the yellow color from the green color in the test vessel is a limitation of the test. However, this difficulty only appeared in 4 of 200 cases, which could not have changed the outcome of our study. In order to make this clear for the readers, we have added this information in the Discussion section of our revised manuscript (page 6–7, lines 235–238).

Reviewer 3, Comment 2:

Minor Critique: Line 216-217, “this study was distinctly different from previously published study”. Please explain the difference clearly and provide the reference for those studies.

Response to Reviewer 3, Comment 2:

Thank you for this comment. Previous studies evaluated the OSOM® BVBLUE® Test for symptomatic, non-pregnant patients [1,2]. In contrast, we have evaluated the point-of-care test for asymptomatic pregnant women, which could have an implication as a potential screening tool during antenatal care. The respective literature is discussed in the paragraph above the one that is mentioned by Reviewer 3 (page 6, lines 221–228). In order to make this point clear for the readers, we have adapted the paragraph as follows: “Our study is distinctly different from previously published studies as we evaluated this test for screening asymptomatic, pregnant women.” (page 6, lines 229–230).

--

REFERENCES

  1. Myziuk, L.; Romanowski, B.; Johnson, S.C. BVBlue test for diagnosis of bacterial vaginosis. J Clin Microbiol 2003, 41, 1925-1928, doi:10.1128/jcm.41.5.1925-1928.2003.
  2. Bradshaw, C.S.; Morton, A.N.; Garland, S.M.; Horvath, L.B.; Kuzevska, I.; Fairley, C.K. Evaluation of a point-of-care test, BVBlue, and clinical and laboratory criteria for diagnosis of bacterial vaginosis. J Clin Microbiol 2005, 43, 1304-1308, doi:10.1128/jcm.43.3.1304-1308.2005.
  3. Vardar, E.; Maral, I.; Inal, M.; Ozgüder, O.; Tasli, F.; Postaci, H. Comparison of Gram stain and Pap smear procedures in the diagnosis of bacterial vaginosis. Infect Dis Obstet Gynecol 2002, 10, 203-207, doi:10.1155/s1064744902000236.
  4. Ying, G.S.; Maguire, M.G.; Glynn, R.J.; Rosner, B. Calculating Sensitivity, Specificity, and Predictive Values for Correlated Eye Data. Invest Ophthalmol Vis Sci 2020, 61, 29, doi:10.1167/iovs.61.11.29.
  5. Farr, A.; Kiss, H.; Hagmann, M.; Marschalek, J.; Husslein, P.; Petricevic, L. Routine Use of an Antenatal Infection Screen-and-Treat Program to Prevent Preterm Birth: Long-Term Experience at a Tertiary Referral Center. Birth (Berkeley, Calif.) 2015, 42, 173-180, doi:10.1111/birt.12154.
  6. Foessleitner, P.; Kiss, H.; Deinsberger, J.; Ott, J.; Zierhut, L.; Farr, A. Validation of the SavvyCheck™ Vaginal Yeast Test for Screening Pregnant Women for Vulvovaginal Candidosis: A Prospective, Cross-Sectional Study. J Fungi (Basel) 2021, 7, doi:10.3390/jof7030233.
  7. Hoyme, U.B.; Saling, E. Efficient prematurity prevention is possible by pH-self measurement and immediate therapy of threatening ascending infection. Eur J Obstet Gynecol Reprod Biol 2004, 115, 148-153, doi:10.1016/j.ejogrb.2004.02.038.
  8. Akobeng, A.K. Understanding diagnostic tests 1: sensitivity, specificity and predictive values. Acta Paediatr 2007, 96, 338-341, doi:10.1111/j.1651-2227.2006.00180.x.
  9. Trevethan, R. Sensitivity, Specificity, and Predictive Values: Foundations, Pliabilities, and Pitfalls in Research and Practice. Front Public Health 2017, 5, 307, doi:10.3389/fpubh.2017.00307.
  10. Romero, R.; Espinoza, J.; Kusanovic, J.P.; Gotsch, F.; Hassan, S.; Erez, O.; Chaiworapongsa, T.; Mazor, M. The preterm parturition syndrome. Bjog 2006, 113 Suppl 3, 17-42, doi:10.1111/j.1471-0528.2006.01120.x.
  11. Lamont, R.F.; Nhan-Chang, C.L.; Sobel, J.D.; Workowski, K.; Conde-Agudelo, A.; Romero, R. Treatment of abnormal vaginal flora in early pregnancy with clindamycin for the prevention of spontaneous preterm birth: a systematic review and metaanalysis. Am J Obstet Gynecol 2011, 205, 177-190, doi:10.1016/j.ajog.2011.03.047.

Round 2

Reviewer 1 Report

Even after the author's response, the manuscript suffers from limitations that justify the manuscript to be rejected in the present form.

  • The determination of the number of patients to include must be based on the expected difference, the statistical parameters, etc. and not on local parameters as "(i) the patient volume at our department, (ii) the prevalence of BV within our patient collective, (iii) the planned recruitment period". After its determination, if the number of patients could not be achieved, the study design has to be modified. It is important for the reader to obtain such data, in order to evaluate the quality of the study.
  • Even if the authors refer to publications that demonstrate that Gram-staining could be considered as a gold standard, they have to be aware of the inter-operator variability. When considering subjective parameters as microscopy examination, the authors have to be aware of this risk and consider it in the design of their study. 
  • Using an external prevalence will risk to bias evaluation of the PPV and NPV. The case-control design will not allow this determination, as the prevalence is fixed by the investigators. I think that, if the authors want to determine PPV/NPV, they have to determine the prevalence in their cohort (a different set of patient). Personnaly, I would recommend to determine Likelihood ratios that are prevalence-independent and allow robust comparison between studies.

Author Response

REVIEWER 1

Reviewer 1, Comment 1:

The determination of the number of patients to include must be based on the expected difference, the statistical parameters, etc. and not on local parameters as "(i) the patient volume at our department, (ii) the prevalence of BV within our patient collective, (iii) the planned recruitment period". After its determination, if the number of patients could not be achieved, the study design has to be modified. It is important for the reader to obtain such data, in order to evaluate the quality of the study.

Response to Reviewer 1, Comment 1:

We thank Reviewer 1 for the valuable feedback. Scientists should report the true circumstances of their research, which is why we stated that the number of included study participants was considering the patient volume at our department, the prevalence of BV within our patient collective, the planned recruitment period, as well as the number of participants in the available literature for non-pregnant women [1,2]. However, we understand that this justification does not meet the criteria of a power-analysis, which is why we now performed a post-hoc calculation of the sample size, which was needed to screen in order to achieve accurate sensitivity and specificity for our study. By using the post-hoc calculation method as described by Hajian-Tilaki [3], we found that we would needed to screen a number of 643 cases in order to achieve adequate and reliable results. In fact, we have screened 1,972 pregnant women during the study period, which is why we believe that our results are accurate.

Reviewer 1, Comment 2:

Even if the authors refer to publications that demonstrate that Gram-staining could be considered as a gold standard, they have to be aware of the inter-operator variability. When considering subjective parameters as microscopy examination, the authors have to be aware of this risk and consider it in the design of their study.

Response to Reviewer 1, Comment 2:

We thank Reviewer 1 for this comment. Indeed, we are aware of the inter-operator variability of Gram staining. However, the inter-operator variability in our study was likely small, as Gram staining procedure and microscopic analyses were performed by a small group of homogeneously trained and experienced biomedical laboratory assistants that are specialized in gynecological cytopathology at our ISO-certified laboratory. Our biomedical laboratory assistants evaluate approximately 2,500 Gram-stained smears per year (500–1,000 per assistant), and we therefore consider them as accurate. However, in order to adequately respond to the comment of Reviewer 1, we have now included the inter-operator variability as a potential limitation of our study (page 7, lines 254–255).

Apart from that, a validation study needs to compare the method of interest with the gold standard method. We decided to use Gram stain, as it is widely accepted as gold standard in the diagnosis of BV [4]. The alternative option would have been to compare the point-of-care test with culture methods, which is, however, considered inadequate for the diagnosis of BV, since the detection of Gardnerella vaginalis on culture methods does not necessarily indicate BV [5]. In fact, Gardnerella vaginalis is also part of the physiological lactobacilli-dominated microbiota of healthy women, albeit in smaller numbers.

Reviewer 1, Comment 3:

Using an external prevalence will risk to bias evaluation of the PPV and NPV. The case-control design will not allow this determination, as the prevalence is fixed by the investigators. I think that, if the authors want to determine PPV/NPV, they have to determine the prevalence in their cohort (a different set of patient). Personally, I would recommend to determine Likelihood ratios that are prevalence-independent and allow robust comparison between studies.

Response to Reviewer 1, Comment 3:

Thank you for this comment. As requested by both reviewers (also see Response to Reviewer 2, Comment 1), we have now retrospectively collected the entire data of our screened cohort, calculating the prevalence of BV within this cohort. In fact, we found a slightly higher BV prevalence as initially suggested (9.2%) during the study period. Our statistician therefore re-calculated PPV and NPV, showing that the PPV remained 100%, and that the NPV changed from 98.5% to 98.1%. In the revised version of our manuscript, we have now updated all numbers within the mainbody and tables. Revised Table 2 is shown below. The use of likelihood ratios as an alternative became therefore obsolete, as we now analyzed our entire data. Of note, the conclusion of our study remained the same considering these revised results.

Table 2. Validation of the OSOM® BVBLUE® Test in comparison to Gram staining in 200 asymptomatic pregnant women screened for BV.

Parameter

Study group

(n = 100)

Control group

(n = 100)

Total

(n = 200)

Positive test

81

0

81

Negative test

19

100

119

Total

100

100

200

Sensitivity = 81.0%, specificity = 100.0%, positive predictive value = 100.0%, negative predictive value = 98.1%; prevalence: 9.2%. BV, bacterial vaginosis.

Reviewer 2 Report

Before replying to any of the authors' responses or edits, I want to point out that the review system does not allow me to see their responses to reviewer one. This unfortunately means that I might be missing important details in their responses where they refer to them. 

Foessleitner et al., have appropriately responded to the majority of my comments, but there are still two important points that should be addressed:

The most important part is the definition of when participants are included or excluded from a study. The authors argue that they invite participants to participate in the study after they have passed the Gram staining inclusion criteria. I do not find that to be an acceptable way to report their recruitment. The authors must report how many women were considered for participation, how many were ineligible based on criteria in section 2.2, and how many were excluded based on their Nugent score. To address this, the authors should report the number of women in the following categories:

  • Women considered for participation
  • Ineligible women (due to criteria in section 2.2)
  • Women excluded due to intermediate BV (Nugent score 4-6)
  • Women excluded with no BV but Candida
  • Women not invited with no BV (Nugent score 0-3)

While I understand that this information is not the aim of the study it must be reported. 

The second concern general prevalence of BV. The optimal solution would be to use the information requested above to find the actual prevalence of BV for this population. If the authors maintain that they will use the prevalence reported by Farr et al., then I would like the authors to explain why they use the prevalence of "BV (alone)" (7.4%), without including the additional 1.3% that has both "BV and Candida", which would give an overall BV prevalence of 8.7%

Lastly, I have two minor points:

Please add the statistical test used to the Table 1 legend.

In line 95: To me, the phrasing "... at a laboratory certified ..." makes it sound like the samples were sent to an external laboratory for analysis, while the response from the authors makes it clear that they are referring to their own laboratory. I suggest changing the sentence to "... at our laboratory, which is certified ..."  

Author Response

REVIEWER 2

Reviewer 2, Comment 1:

Foessleitner et al., have appropriately responded to the majority of my comments, but there are still two important points that should be addressed:

The most important part is the definition of when participants are included or excluded from a study. The authors argue that they invite participants to participate in the study after they have passed the Gram staining inclusion criteria. I do not find that to be an acceptable way to report their recruitment. The authors must report how many women were considered for participation, how many were ineligible based on criteria in section 2.2, and how many were excluded based on their Nugent score. To address this, the authors should report the number of women in the following categories:

  • Women considered for participation
  • Ineligible women (due to criteria in section 2.2)
  • Women excluded due to intermediate BV (Nugent score 4-6)
  • Women excluded with no BV but Candida
  • Women not invited with no BV (Nugent score 0-3)

While I understand that this information is not the aim of the study it must be reported.

Response to Reviewer 2, Comment 1:

We thank Reviewer 2 for this useful comment. We have now reported all numbers in detail as follows: In total, 1,972 pregnant women underwent antenatal infection screening during the study period.  Out of these, 1,473 women (74.7%) had a normal vaginal microbiota on Gram stain, 318 women (16.1%) had dysbiosis, and 181 women (9.2%) had BV. For validating the point-of-care test, we had to exclude the 318 women with dysbiosis (i.e., Nugent score 4–6), as well as another 81 women who reported ongoing or recent symptoms, or who had recently either received antibiotic treatment or any vaginal medication according to our protocol. Another 165 women with a normal microbiota (i.e., Nugent score 0–3) who were found to have vulvovaginal candidosis were excluded from the study. Out of the remaining 1,408 asymptomatic pregnant women, 100 were diagnosed with BV and 1,308 women had a normal microbiota without VVC. The 100 women with BV as well as 100 out of the 1,308 women with normal microbiota were invited to participate in our study and all women gave their consent to be participate. In order to make this clear for the readers, we have now revised Figure 1 (page 4 of the manuscript). Consequently, we added a paragraph to the Results section, describing these numbers (page 4, lines 146–154).

Reviewer 2, Comment 2:

The second concern general prevalence of BV. The optimal solution would be to use the information requested above to find the actual prevalence of BV for this population. If the authors maintain that they will use the prevalence reported by Farr et al., then I would like the authors to explain why they use the prevalence of "BV (alone)" (7.4%), without including the additional 1.3% that has both "BV and Candida", which would give an overall BV prevalence of 8.7%

Response to Reviewer 2, Comment 2:

Please see our Response to Reviewer 2, Comment 1. We managed to retrospectively collect the entire screening data and have therefore re-calculated the results of our study. However, the conclusion remained the same.

Reviewer 2, Comment 3:

Please add the statistical test used to the Table 1 legend.

Response to Reviewer 2, Comment 3:

Thank you for this important point. The statistical tests used to calculate the p-values were now added to the legend of Table 1 and to the Methods section of our revised manuscript (page 3, lines 139–144).

Reviewer 2, Comment 4:

In line 95: To me, the phrasing "... at a laboratory certified ..." makes it sound like the samples were sent to an external laboratory for analysis, while the response from the authors makes it clear that they are referring to their own laboratory. I suggest changing the sentence to "... at our laboratory, which is certified ..." 

Response to Reviewer 2, Comment 4:

We thank Reviewer 2 for this suggestion. We have rephrased the respective paragraph as suggested (pages 2–3, lines 93–96).
